# FAIRNESS-AWARE ATTENTION FOR CONTRASTIVE LEARNING

## ABSTRACT

Contrastive learning has proven instrumental in learning unbiased representations of data, especially in complex environments characterized by high-cardinality and high-dimensional sensitive information. However, existing approaches within this setting require predefined modelling assumptions of bias-causing interactions that limit the model's ability to learn debiased representations. In this work, we propose a new method for fair contrastive learning that employs an attention mechanism to model bias-causing interactions, enabling the learning of a fairer and semantically richer embedding space. In particular, our attention mechanism avoids bias-causing samples that confound the model and focuses on bias-reducing samples that help learn semantically meaningful representations. We verify the advantages of our method against existing baselines in fair contrastive learning and show that our approach can significantly boost bias removal from learned representations without compromising downstream accuracy.

## 1 INTRODUCTION

Machine learning models are continuing to achieve impressive results across diverse domains. Wider adoption and development of such models pose immense opportunity, yet there simultaneously exists a substantial risk of societal harm in situations where models propagate forward biases encoded in training data (Lv et al., 2023; Creager et al., 2019; Madras et al., 2018). In particular, existing facial recognition systems demonstrate racial bias in their classifications, failing to recognize people from certain ethnic groups (Cavazos et al., 2020). In addition, generative language models, such as GPT-2, have been shown to reproduce gender bias in their generated text, for example in systematically assuming doctors are male and nurses are female among other socially biased outcomes (Kirk et al., 2021; Bender et al., 2021).

One effective approach to resolving this problem is fair representation learning (Wang et al. (2019); Khajehnejad et al. (2022); Zhang et al. (2023)). This approach recognizes that bias is encoded at the data level and looks to learn representations of the data that preserve relevant semantic content while removing sensitive information related to a specified protected attribute, such as race, gender, age, geographic location, and so on. Specifically, contrastive learning has been used to learn fair representations. This technique learns similar representations for *positively-paired* samples and dissimilar representations for *negatively-paired* samples (Chuang et al. (2020); Tian et al. (2020); He et al. (2020)). For example, a positive-pair in the vision setting may be augmentations of the same image and a negative-pair may be any pair of distinct images (Chen et al., 2020). Thus, designing positive and negative pairs in the right way informs the model what features are semantically meaningful and what features are irrelevant in distinguishing samples. This approach then lends itself to fairness when we design positives and negatives such that the model learns representations that are invariant to the protected attribute, thereby removing sensitive information related to the protected attribute from the learned representations.

Existing work in fair contrastive learning often assumes the protected attribute to be a binary variable, such as gender or minority status. Popular fair contrastive learning methods include bias-label flipping, bias-label augmenting, and parity-enforcing regularizers (Cheng et al., 2021; Ling et al., 2022; Zhang et al., 2022; Shen et al., 2021; Barbano et al., 2022; Cheng et al., 2021; Hong & Yang, 2021). These approaches, while effective in the binary setting, are limited in their usability due to their conceptualization of fairness as a binary problem in which samples can only belong to one of two groups in terms of the protected attribute, such as male/female or majority/minority. As a

result, they fail to generalize to the harder and more general problem setting of high-cardinality, high-dimensional, and/or continuous protected attributes. Recently, Tsai et al. (2022) considers the continuous protected attribute setting and proposes a conditional sampling procedure in which negative pairs are sampled according to their similarity in the bias dimension. This approach, however, requires a pre-defined kernel function which imposes strong assumptions on the bias-causing interactions among samples. This is because the chosen kernel function specifies exactly for any given similarity between negative samples in the bias dimension the relevance of that sample for contrasting with the positive pair. These strong assumptions on the bias-causing interactions among samples limits the model's ability to learn fair representations and additionally requires expensive matrix inversion operations.

**Contribution:** We propose the **Fa**irness-Awa**re** (FARE) attention mechanism that attends towards bias-reducing samples and avoids bias-causing samples that confound the model. We further leverage sparsification via locality-sensitive hashing (Shrivastava & Li, 2014; Andoni et al., 2015; Kitaev et al., 2020) to discard extreme bias-causing samples in FARE and propose the Sparse **Fa**irness-Awa**re** (SparseFARE) attention. Our approach is based on the assumption that using similar samples in the bias dimension should prevent the protected information from being used to differentiate samples, thereby removing the sensitive information from the learned representations. FARE and SparseFARE are designed to learn a similarity metric across the protected attributes that capture the bias-causing interactions. To train FARE, we derive the new Fair Attention-Contrastive (FARE-Contrast) loss that expresses the negative samples as the output of the FARE attention mechanism, in which similarity scores of negative samples are conditioned by learned attention scores. Our contribution is three-fold.

- We develop FARE, a novel fairness-aware attention mechanism that captures the bias-causing interactions to reduce bias and learn semantically relevant embedding spaces.
- We sparsify FARE to enhance its ability to learn fair representation by discarding extreme bias-causing samples, resulting in the SparseFARE attention.
- We derive the FAREContrast loss to train FARE.

We empirically demonstrate that compared to the baseline methods, FARE alleviates a significantly larger amount of bias without compromising downstream accuracy and with lower computational complexity.

**Notation:** Let calligraphic letters represent dataspaces (e.g $\mathcal{X}$), capital letters represent random variables (e.g $X$), lower case letter represent their outcomes (e.g $x$), and $P$. represent distributions of the variable in the subscript (e.g $P_X$). We abuse notation slightly and also denote matrices by capital letters and vectors comprising matrices by lower case letters (e.g $Q = [q_1, \ldots, q_n]^\top$ where $Q \in \mathbb{R}^{n \times k}$ and $q_i \in \mathbb{R}^k$), in which cases we make clear that the capital and lower case letters correspond to matrices and vectors rather than random variables and outcomes.

**Organization:** We structure this paper as follows: Section 2 establishes the necessary technical background. Section 3 derives the FARE and SparseFARE attention mechanisms, as well as the FAREContrast objective loss. Section 4 provides the empirical validation of our proposed attention-based methods. Section 5 discusses related work. The paper ends with concluding remarks. Additional details on experimental setup, further results, and other technical details are found in the Appendix.

## 2 BACKGROUND

In this section, we summarize the technical preliminaries needed to develop our method, comprising conditional contrastive learning and attention mechanisms.

### 2.1 CONDITIONAL CONTRASTIVE LEARNING

Contrastive methods learn an encoding of the data such that similar samples are near each other while dissimilar samples are far from each other in the embedding space (Chen et al., 2020; He et al., 2020; Hjelm et al., 2018). This is done by sampling a positive sample $y_{pos}$ and negative sample $y_{neg}$ for any given $x \in \mathcal{X}$, where the encoder learns a representation such that $x$ and $y_{pos}$ are near each other while $x$ and $y_{neg}$ are distant. Conditional contrastive methods extend this approach to allow for conditional sampling on an additional variable $Z$, which in the fairness setting is a protected attribute (Tsai et al., 2022). In particular, the data pair $(x, y_{pos})$ is sampled from $P_{XY|Z=z}$

as $x$ and $y_{pos}$ are views of one another (obtained via augmentation) and $(x, y_{neg})$ is sampled from $P_{X|Z=z}P_{Y|Z=z}$ as $x$ and $y_{neg}$ are two distinct samples (Oord et al., 2018; Tsai et al., 2021a).

The Fair-InfoNCE objective (Tsai et al. (2021b)) is then defined as:

$$\sup_f \mathbb{E}_{z \sim P_Z, \ (x, y_{pos}) \sim P_{XY|Z=z}, \ \{y_{neg}\}_{i=1}^b \sim P_{Y|Z=z}^{\otimes b}} \left[ \log \frac{e^{f(x, y_{pos})}}{e^{f(x, y_{pos})} + \sum_{i=1}^b e^{f(x, y_{neg,i})}} \right] \quad (1)$$

where $b$ denotes the batch size and $f : \mathcal{X} \times \mathcal{Y} \to \mathbb{R}$ is a mapping parameterized by neural networks $g_{\theta_X}, g_{\theta_Y}$, given by:

$$f(x, y) = \text{cosine similarity}\Big(g_{\theta_X}(x), g_{\theta_Y}(y)\Big) / \tau, \quad (2)$$

where the networks are themselves parameterized by $\theta_X, \theta_Y$ and $\tau$ is a hyperparameter scaling the cosine similarity (Chen et al., 2020). In many cases, as in ours, $g_{\theta_X} = g_{\theta_Y}$. The function $f$ from 2 is referred to as the scoring function between samples $x, y$ and evaluates the similarity between the learned embeddings of the neural network. Hence, the learning objective aims to maximize the score for positive pairs and minimize the score for negative pairs.

We also express the exponential scoring function in terms of an inner product in a Reproducing Kernel Hilbert Space (RKHS) with corresponding feature map (Tsai et al., 2022) as follows:

$$e^{f(x,y)} = \exp\Big(\text{cosine similarity}(g_{\theta_X}(x), g_{\theta_Y}(y)) / \tau\Big) := \Big\langle \phi(g_{\theta_X}(x)), \phi(g_{\theta_Y}(y)) \Big\rangle_{\mathcal{H}}, \quad (3)$$

where $\langle \cdot, \cdot \rangle_{\mathcal{H}}$ is the inner product in RKHS $\mathcal{H}$ and $\phi$ is the feature map associated with $\mathcal{H}$.[1]

## 2.2 ATTENTION MECHANISM

The scaled dot-product attention mechanism (Vaswani et al. (2017)) is given as:

$$\text{Attention}(Q, K, V) = \underbrace{\text{softmax}\left(\frac{QK^\top}{\rho}\right)}_{P} V,$$

where $Q = TW_Q$, $K = SW_K$ and $V = UW_V$ representing the queries, keys and values respectively, which are obtained via learnable linear projections, $W_Q, W_K \in \mathbb{R}^{d_m \times d_k}, W_V \in \mathbb{R}^{d_m \times d_v}$, of data matrices $S \in \mathbb{R}^{n \times d_m}, T \in \mathbb{R}^{n \times d_m}, U \in \mathbb{R}^{n \times d_m}$ where $n$ is the sequence length, $d_m$ is the embedding dimension and $d_v$ is the chosen hidden dimension of the projection subspaces. The softmax operator is applied row-wise, and $\rho$ is a temperature hyperparameter most often set to $\sqrt{d}$. We refer to $P \in \mathbb{R}^{n \times n}$ as the attention map, which contains information regarding the learned similarities between individual keys and queries. In many cases, $S = T = U$, referred to as self-attention. Our model is inspired by self-attention, where we take $S = T = Z$, where $Z = [z_1, \ldots, z_n]^\top$ is the input sequence of protected attributes, but $U \neq Z$. Instead, for our purposes, we take $U \in \mathbb{R}^{n \times n}$ with entries $[U]_{ij} = e^{f(x_i, y_j)}$ which is the matrix of similarity scores between samples $x_i, y_j$. Furthermore, we pass this matrix straight into the attention computation without projecting it with $W_V$, and so $U = V$ (see Remark 2 in section 3.1). This is because we wish to use the attention map $P$ to provide contextual information to condition the similarity scores, $e^{f(x_i, y_j)}$, rather than the sensitive attributes. Under this setup, the attention score $p_{ij}$ and the output $o_i$ of the attention as follows:

$$p_{ij} = \text{softmax}((W_Q t_i)^\top (W_K s_j) / \rho), \ \ o_i = \sum_j^n p_{ij} e^{f(x_i, y_j)}. \quad (4)$$

The output of the attention mechanism can therefore be interpreted as a conditionally weighted sum over the values with weights provided by the attention scores. Section 3 illustrates how these attention scores capture bias-causing interactions, and so the attention outputs are equivalently the values conditioned by their bias-causing potential, which serves the purpose of accentuating bias-reducing samples and attenuating bias-causing samples, which helps to learn debiased representations.

---

[1]Note the exponential of the scoring function is a proper kernel, as the scoring function is a proper kernel and the exponential of proper kernels are proper kernels as well.

## 3 FAIRNESS MEETS ATTENTION

In this section, we present our **Fa**irness-Awa**re** (FARE) attention mechanism. FARE focuses on negative samples to reduce bias and improve the semantic content of the learned representations. The model passes the negative samples through an attention mechanism where the outputs are the linearly weighted sum of negative samples according to their bias dimension and their semantic relevance, where the weights are the attention scores (see Section 3.1). The attention matrix is then sparsified such that high bias-inducing samples are given zero attention scores (see Section 3.2), resulting in the Sparse **Fa**irness-Awa**re** attention (SparseFARE). FARE and SparseFARE are trained to minimize a novel Fair Attention-Contrastive (FAREContrast) loss, which incorporates FARE/SparseFARE into the Fair-InfoNCE objective in Eqn. 21 FAREContrast loss allows FARE-based methods to capture the bias-causing interactions over samples while learning good representation for downstream tasks.

### 3.1 FARE: FAIRNESS-AWARE ATTENTION

The only available data is the batch of triplets $\{x_i, y_i, z_i\}_{i=1}^b$, which are independently sampled from the joint distribution $P_{XYZ}^{\otimes b}$ with $b$ being the batch size, and we do not have access to data pairs from the conditional distribution $P_{X|Z}P_{Y|Z}$. Therefore, we aim to bypass the conditional sampling process from the Fair-InfoNCE objective in Eqn. 21. In particular, to transform the Fair-InfoNCE objective into an alternative version that does not require conditional sampling, we estimate the scoring function $e^{f(x,y)}$ for $(x, y) \sim P_{X|Z}P_{Y|Z}$ in Eqn. 21 given only $\{x_i, y_i, z_i\}_{i=1}^b \sim P_{XYZ}^{\otimes b}$. We do this by employing kernel density estimators to view the desired similarity score as the output of an attention mechanism, which leverages attention as kernelized non-linear similiarity score (Tsai et al., 2019; Parzen, 1962; Rosenblatt, 1956). *Given an anchor $(x_i, z_i)$, FARE estimates the similarity score between $x_i$ and $y \sim P_{Y|Z=z_i}$ by conditionally weighting all samples in the batch, with weights provided by learned attention scores over the protected attributes.* We derive FARE below.

For any $Z = z$, given $y \sim P_{Y|Z=z}$, we estimate $\phi(g_{\theta_Y}(y))$ by $\mathbb{E}_{y \sim P_{Y|Z=z}}[\phi(g_{\theta_Y}(y))]$ as follows:

$$\phi(g_{\theta_Y}(y)) \approx \mathbb{E}_{y \sim P_{Y|Z=z}}[\phi(g_{\theta_Y}(y))] = \int \phi(g_{\theta_Y}(y))P(y|z)dy = \int \phi(g_{\theta_Y}(y))\frac{P(y,z)}{P(z)}dy. \quad (5)$$

We then plug Eqn. 5 into Eqn. 2 for the data pair $(x_i, z_i)$ to estimate $e^{f(x_i,y)}$ when $y \sim P_{Y|Z=z_i}$ as

$$\begin{aligned}
\hat{e}_{\text{conditioned}}^{f(x_i,y)} &\approx \left\langle \phi(g_{\theta_X}(x_i)), \int \phi(g_{\theta_Y}(y))\frac{P(y,z)}{P(z)}dy \right\rangle_{\mathcal{H}} \\
&= \text{tr}\left( \phi(g_{\theta_X}(x_i))^\top \int \phi(g_{\theta_Y}(y))\frac{P(y,z)}{P(z)}dy \right) \\
&= \phi(g_{\theta_X}(x_i))^\top \int \phi(g_{\theta_Y}(y))\frac{P(y,z)}{P(z)}dy.
\end{aligned} \quad (6)$$

Here we denote the conditional estimation of the scoring function $e^{f(x,y)}$ for $(x, y) \sim P_{X|Z}P_{Y|Z}$ by $\hat{e}_{\text{conditioned}}^{f(x,y)}$.

**Kernel density estimator.** To estimate $P(y, z)$ and $P(z)$, we employ the kernel density estimation approach (Parzen, 1962; Rosenblatt, 1956). In particular, by using the isotropic Gaussian kernel with bandwidth $\sigma$, we obtain the following estimators of $P(y, z)$ and $P(z)$:

$$\hat{P}_\sigma(y,z) = \frac{1}{b}\sum_{j=1}^b \varphi_\sigma(y-y_j)\varphi_\sigma(z-z_j), \ \ \hat{P}_\sigma(z) = \frac{1}{b}\sum_{j=1}^b \varphi_\sigma(z-z_j), \quad (7)$$

where $\varphi_\sigma(\cdot)$ is the isotropic multivariate Gaussian density function with diagonal covariance matrix $\sigma^2\mathbf{I}$. Given Eqn. 6 and the kernel density estimators in Eqns. 7, we attain the following conditional

estimation of the scoring function:

$$
\begin{aligned}
\hat{e}^{f(x_i,y)}_{\text{conditioned}} &= \phi(g_{\theta_X}(x_i))^\top \int \phi(g_{\theta_Y}(y)) \frac{\hat{P}(y,z)}{\hat{P}(z)} dy \\
&= \phi(g_{\theta_X}(x_i))^\top \int \phi(g_{\theta_Y}(y)) \frac{\sum_{j=1}^b \varphi_\sigma(y-y_j)\varphi_\sigma(z-z_j)}{\sum_{j=1}^b \varphi_\sigma(z-z_j)} dy \\
&= \phi(g_{\theta_X}(x_i))^\top \frac{\sum_{j=1}^b \varphi_\sigma(z-z_j) \int \phi(g_{\theta_Y}(y))\varphi_\sigma(y-y_j) dy}{\sum_{j=1}^b \varphi_\sigma(z-z_j)} \\
&= \frac{\sum_{j=1}^b \left[\phi(g_{\theta_X}(x_i))^\top \phi(g_{\theta_Y}(y_j))\right] \varphi_\sigma(z-z_j)}{\sum_{j=1}^b \varphi_\sigma(z-z_j)}.
\end{aligned}
\tag{8}
$$

**Connection to Attention Mechanism.** In Eqn. 8, we replace $\varphi_\sigma$ by the formula of the isotropic multivariate Gaussian density function with diagonal covariance matrix $\sigma^2\mathbf{I}$ and obtain

$$
\begin{aligned}
\hat{e}^{f(x_i,y)}_{\text{conditioned}} &= \frac{\sum_{j=1}^b \left[\phi(g_{\theta_X}(x_i))^\top \phi(g_{\theta_Y}(y_j))\right] \exp\left(-\|z-z_j\|^2/2\sigma^2\right)}{\sum_{j=1}^b \exp\left(-\|z-z_j\|^2/2\sigma^2\right)} \\
&= \frac{\sum_{j=1}^b \left[\phi(g_{\theta_X}(x_i))^\top \phi(g_{\theta_Y}(y_j))\right] \exp\left(-(\|z\|^2+\|z_j\|^2)/2\sigma^2\right) \exp\left(z^\top z_j/\sigma^2\right)}{\sum_{j=1}^b \exp\left(-(\|z\|^2+\|z_j\|^2)/2\sigma^2\right) \exp\left(z^\top z_j/\sigma^2\right)}
\end{aligned}
\tag{9}
$$

If we further assume that $z_j$, $j = 1, 2, \ldots, b$ are normalized and choose $\sigma^2 = \rho$, where $\rho$ is the attention temperature hyperparameter in Eqn. 4, the conditionally estimated scoring function is then

$$
\begin{aligned}
\hat{e}^{f(x_i,y)}_{\text{conditioned}} &= \frac{\sum_{j=1}^b \left[\phi(g_{\theta_X}(x_i))^\top \phi(g_{\theta_Y}(y_j))\right] \exp\left(z^\top z_j/\rho\right)}{\sum_{j=1}^b \exp\left(z^\top z_j/\rho\right)} \\
&= \sum_{j=1}^b \text{softmax}\left(z^\top z_j/\rho\right) \left[\phi(g_{\theta_X}(x_i))^\top \phi(g_{\theta_Y}(y_j))\right].
\end{aligned}
\tag{10}
$$

Plugging in the observed outcome of the protected attribute, $z = z_i$, and allowing $z_i$ and $z_j$ to be transformed by learnable linear transformation, $W_Q, W_K$, the conditionally estimated similarity score $e^{f(x_i,y)}$ when $y \sim P_{Y|Z=z_i}$ is then given by

$$
\hat{e}^{f(x_i,y)}_{\text{conditioned}} = \sum_{j=1}^b \underbrace{\text{softmax}\left((W_Q z_i)^\top W_K z_j/\rho\right)}_{p_{ij}} \underbrace{\left[\phi(g_{\theta_X}(x_i))^\top \phi(g_{\theta_Y}(y_j))\right]}_{e^{f(x_i,y_j)}},
\tag{11}
$$

which is the output of an attention mechanism with values given by the unconditioned similarity scores between samples, $e^{f(x_i,y_j)}$, and attention scores $p_{ij}$ computed over the protected attributes $(z_i, z_j)$. Thus, the similarity scoring function estimation between $x_i$ and $y \sim P_{Y|Z=z_i}$ can be approximated by an attention output. We summarize this new result in the following proposition.

**Proposition 1** (Conditional Estimation of $e^{f(x_i,y)}$ when $y \sim P_{Y|Z=z_i}$). *Given* $\{x_i, y_i, z_i\}_{i=1}^b \sim P^{\otimes b}_{XYZ}$, *the finite-sample estimation of* $e^{f(x_i,y)}$ *is* $\sum_{j=1}^b \text{softmax}\left((W_Q z_i)^\top W_K z_j/\rho\right) \left[\phi(g_{\theta_X}(x_i))^\top \phi(g_{\theta_Y}(y_j))\right]$, *which is the output of an attention mechanism.*

Hence, the attention scores $p_{ij}$ condition the similarity scores $e^{f(x_i,y_j)}$, i.e., for any data pair $(x_i, y_j)$, their similarity is accentuated/attenuated depending on the attention between the protected attributes $(z_i, z_j)$. At a high level, when $z_i$ is dissimilar from $z_j$, $x_j$ is likely to cause a bias in the learned representations, and we expect the attention mechanism to divert its focus from that sample. Conversely, when $z_i$ is similar to $z_j$, $x_j$ is likely to reduce the bias in the learned representations, and the attention mechanism should place more focus on that sample. However, rather than specifying

the extent to which similarities over the bias dimension should mask out samples via a pre-defined kernel as in Tsai et al. (2022), we allow the attention mechanism to learn this metric given the task. This flexibility allows the model to focus on samples that are bias-reducing and shift its emphasis away from samples that are bias-causing while simultaneously adapting towards the overall task of learning semantically meaningful representations.

We are now ready to give a full definition of FARE.

**Definition 1** (Fairness-Aware Attention). *Fairness-aware attention (FARE) is an attention mechanism that computes the finite-sample estimation of the similarity scores $e^{f(x_i, y)}$ when $y \sim P_{Y|Z=z_i}$ for $i = 1, 2, \ldots, b$ with $b$ being the batch size. Given $\{(x_i, y_i, z_i)\}_{i=1}^{b} \sim P_{XYZ}^{\otimes b}$, FARE is defined as*

$$FARE(\{(x_i, y_i, z_i)\}_{i=1}^{b}) = \hat{e}_{conditioned}^{f(x_i, y)} = \sum_{j=1}^{b} softmax\left((W_Q z_i)^\top W_K z_j / \rho\right) \left[\phi(g_{\theta_X}(x_i))^\top \phi(g_{\theta_Y}(y_j))\right]$$

(12)

FARE estimates the similarity between any given anchor and negative sample, where the similarity is conditioned according to the protected attribute and the extent to which any sample is likely to bias the representations. By focusing attention on samples according to their bias-inducing characteristics, FARE is able to learn fair representations.

**Remark 1.** *In Proposition 1, the attention score $p_{ij} = softmax\left((W_Q z_i)^\top W_K z_j / \rho\right)$, $i, j = 1, 2, \ldots, b$, provides a context to estimate the similarity score between $x_i$ and $y \sim P_{Y|Z=z_i}$, thus allowing FARE to attain a contextual representation. It has been shown that the ability of the attention mechanism to capture rich and diverse contextual representation is key to the impressive performance of recent deep learning models, including transformers and graph neural networks (Tenney et al., 2019; Vig & Belinkov, 2019; Clark et al., 2019; Voita et al., 2019; Hewitt & Liang, 2019).*

**Remark 2.** *We do not include a learnable value transformation matrix $W_V$ for the values. Rather we pass the unconditioned similarity scores, $e^{f(x_i, y_i)}$, straight into the attention mechanism. This is because a transformation $W_V$ would allow the optimization procedure to take a shortcut and avoid minimizing the objective loss by just sending the value weights to infinity, obtaining 0 loss and thereby preventing the encoder from learning useful representations. More details are given in Appendix E.*

### 3.2 SPARSEFARE: SPARSE FAIRNESS-AWARE ATTENTION

In the previous section, we proposed the use of attention for debiasing representations, we now discuss the role of sparsification towards this goal. If we have prior knowledge on the proportion of samples that need not be considered at all since they are relatively extreme in the bias dimension, then we can discard those samples before computing attention. For example, if color is the protected attribute, then samples with opposing colors such as black/white may be considered extreme in the bias dimension relative to each other. This allows the attention mechanism to be more efficient and debias more aggressively as samples can be given an attention score of exactly 0. We implement the sparse fairness-aware attention (SparseFARE) via locality-sensitive hashing (LSH) (Kitaev et al. (2020)).

**Locality-Sensitive Hashing.** A hashing scheme is locality-sensitive if for all vectors, $z$, assigned hashes $h(z)$, similar vectors are assigned the same hash with high probability and dissimilar vectors are assigned the same hash with low probability (Kitaev et al., 2020). We follow the LSH scheme in (Andoni et al., 2015), which employs random projections $R \in \mathbb{R}^{d_z \times b/2}$ where $[R]_{ij} \sim N(0, 1)$ and assigns hashes by $h(z) = \text{argmax}(\text{concat}(zR, -zR))$.

**Locality-Sensitive Hashing Attention for Fairness.** The basis of the debiasing scheme is the assumption that for anchor $(x_i, z_i)$ and negative sample $(y_j, z_j)$, $y_j$ is likely to increase the bias of the representations when $z_i$ is dissimilar to $z_j$. If we determine some threshold for ignoring $(y_j, z_j)$ when $z_i$ and $z_j$ are sufficiently dissimilar, then we can leverage the LSH scheme to ensure with high probability that $h(y_j) \neq h(x_i)$. Subsequently, if we only permit attention to be calculated within hash buckets (or potentially within hash buckets and across adjacent buckets), we should ignore samples at the relative extremes of $Z$ with high probability to speed up our fairness mechanism and perform more aggressive debiasing by discarding extreme bias-causing samples.

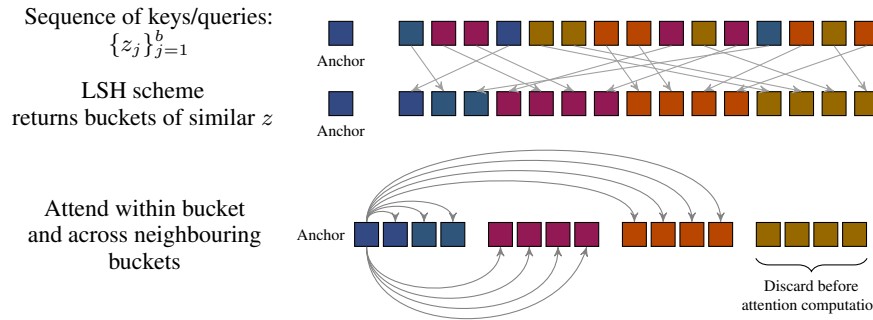

Figure 1: Sparse Fair-Aware Attention (SparseFARE) using LSH to discard bias-causing samples. Relative to the anchor's protected attribute status (blue), the fairness-aware attention (FARE) first groups the samples according to their bias attribute and discards any samples that are likely to be highly bias-inducing (brown). Attention scores between similar and bias-reducing samples are then computed.

For index $i$ of a given query $q_i$, we denote the attention support as $S_i = \{j : h(k_j) = h(k_i)\}$, which is the set of keys hashed to the same bucket and therefore take part in the attention computation with $q_i$.[2] Figure 1 illustrates this scheme.

**SparseFARE Formalization**. Given the LSH scheme for fairness, we now provide the full formulation of the SparseFARE.

**Definition 2** (Fairness-Aware Attention with Sparsification)**.** *Sparse fairness-aware attention (SparseFARE) is a variant of FARE in which the attention map over protected attributes is sparsified by removing entries that are highly bias-inducing. Given $\{(x_i, y_i, z_i)\}_{i=1}^b \sim P_{XYZ}^{\otimes b}$, SparseFARE computes the finite-sample estimation of the similarity scores $e^{f(x_i,y)}$ when $y \sim P_{Y|Z=z_i}$ for $i = 1, 2, \ldots, b$ with $b$ being the batch size as follows:*

$$SparseFARE(\{(x_i, y_i, z_i)\}_{i=1}^b) = \hat{e}_{conditioned}^{f(x_i,y)}$$
$$= \sum_{j \in S_i} softmax\left((W_Q z_i)^\top W_K z_j / \rho - m(j, S_i)\right) \left[\phi(g_{\theta_X}(x_i))^\top \phi(g_{\theta_Y}(y_j))\right],$$

*where $S_i = \{j : h(z_j) = h(z_i)\}$ is the attention support of $i$ and $m(j, S_i) = \begin{cases} \infty & \text{if } j \notin S_i \\ 0 & \text{otherwise} \end{cases}$.*

### 3.3 FAREContrast: Fair Attention-Contrastive Criterion for Contrastive Learning

We now present the Fair Attention-Contrastive (FAREContrast) criterion for fair contrastive learning with FARE. We obtain FAREContrast by replacing the summation over negative samples in the Fair-InfoNCE in Eqn. 21 with the output of FARE. FAREContrast is then defined as

$$\sup_f \mathbb{E}_{\{(x_i,y_i,z_i)\}_{i=1}^b \sim P_{XYZ}^{\otimes b}} \left[\log \frac{e^{f(x_i,y_i)}}{e^{f(x_i,y_i)} + \text{FARE}(\{(x_i, y_i, z_i)\}_{i=1}^b)}\right]. \tag{13}$$

The goal of the FAREContrast criterion is to adapt the Fair-InfoNCE objective such that we avoid conditional sampling. We do this because our FARE attention mechanism avoids conditional sampling of negative pairs by using attention to consider the whole batch and selectively weight samples according to their protected attribute status, in this way focussing on bias-reducing samples for contrasting. Hence we only consider $\{(x_i, y_i, z_i)\}_{i=1}^b \sim P_{XYZ}^{\otimes b}$. Furthermore, we only need FARE for negative samples since only the negatives need to be conditioned for contrasting with the positive pair. The positive pair will necessarily be identical in the bias-dimension as we do not perform augmentations that change the protected attribute. Our method debiases representations by then

---

[2]We denote the attention support as purely intra-bucket here for simplicity. In reality, we will typically allow cross-attention to adjacent buckets as well

| Model | Accuracy (↑) | Bias Removal (↑) |
|---|---|---|
| *Baseline Models* | | |
| InfoNCE (Oord et al., 2018) | $84.1 \pm 1.8$ | $48.8 \pm 4.5$ |
| Fair-InfoNCE (Tsai et al. (2022)) | $85.9 \pm 0.4$ | $64.9 \pm 5.1$ |
| CCLK (Tsai et al. (2022)) | $86.4 \pm 0.9$ | $64.7 \pm 3.9$ |
| *Attention-based Models* | | |
| FARE (**ours**) | $85.7 \pm 0.9$ | $68.4 \pm 4.3$ |
| SparseFARE (**ours**) | $\mathbf{86.4} \pm 1.3$ | $\mathbf{74.0} \pm 3.8$ |

Table 1: Results on colorMNIST. Bias removal is measured by MSE, where high MSE indicates more color information has been removed from the learned representations.

only showing the positive pair negative samples that have similar protected attribute status, such that the protected information is not used to distinguish samples. Hence FAREContrast is obtained by replacing the summation over negative samples in Fair-InfoNCE with FARE.

## 4 EXPERIMENTS

In this section, we numerically justify the advantage of FARE in learning debiased and semantically meaningful representations over the baseline methods including InfoNCE (Oord et al., 2018), Fair-InfoNCE (Tsai et al., 2022), SimCLR (Chen et al., 2020) and the conditional contrastive learning with kernel model (CCLK) (Tsai et al., 2022). We aim to show that: (i) our methods are able to learn representations with sensitive information removed, and (ii) our learned representations maintain relevant semantic content.

**Datasets.** We conduct our experiments on the ColorMNIST dataset (Tsai et al., 2022) and CelebA dataset (Liu et al., 2018). ColorMNIST contains 60,000 handwritten digits with a continuous RGB color randomly assigned to the background of each digit. The color is taken to be the protected attribute. CelebA contains 202,599 images of celebrities with 40 binary annotations indicating hair color, gender, and many other attributes. We take Attractive as target and Young and Male as sensitive attributes simultaneously.

**Evaluation Protocol.** To evaluate representation quality, we adopt the common technique of freezing the encoder and training a linear classifier using the true labels on the encoded representations and measuring accuracy. To evaluate bias removal in the continuous setting of ColorMNIST, we follow the protocol of Tsai et al. (2022) and train a linear layer on the encoded representations to predict each samples' protected attribute. We use the mean squared error (MSE) of predicting the color as a proxy for the extent to which the sensitive information has been removed, where higher loss indicates more sensitive information has been removed. For CelebA, we measure fairness in this binary scenario using the common metric Equalized Odds (Hardt et al., 2016) where a lower score indicates a fairer model. Additional empirical results and experimental details are provided in the Appendix A.

| Model | Acc. (↑) | EO (↓) |
|---|---|---|
| SimCLR (Chen et al., 2020) | **77.7** | 39.6 |
| *Kernel-based Models (Tsai et al., 2022)* | | |
| CCLK-Cosine | 70.2 | 22.4 |
| CCLK-RBF | 69.9 | 21.8 |
| CCLK-Linear | 71.1 | 21.1 |
| CCLK-Polynomial | 71.0 | 20.8 |
| CCLK-Laplacian | 70.0 | 20.8 |
| *Attention-based Models* | | |
| FARE (**ours**) | 73.7 | 23.5 |
| SparseFARE (**ours**) | 70.4 | **18.7** |

Table 2: CelebA Results. Fare and SparseFARE in comparison with kernel baselines under various kernel specifications.

**Results.** Table 1 shows experimental results on the colorMNIST dataset. Our FARE and Sparse-FARE outperform the baseline methods in terms of bias removal while achieving comparable and better top-1 accuracies. In particular, taking accuracy and bias removal together, SparseFARE is able to weakly Pareto dominate all comparative models, learning substantially less biased representations without compromising downstream accuracy.

Table 2 shows the results of the attention-based models on the CelebA dataset. We find that Sim-CLR achieves highest accuracy while SparseFARE achieves the best fairness. Given that Young and Male are both highly correlated with Attractive, it is intuitive that SimCLR attains top accuracy, as SimCLR does not attempt to remove information relating to these two attributes and so is able to leverage the correlation between attributes and target to make more accurate predictions. SparseFARE Pareto dominates all kernel models in terms of fairness and accuracy except for Linear and Polynomial, which achieve marginally higher accuracy. SparseFARE nonetheless attains a better fairness-accuracy tradeoff curve than these two kernels and so for any given level of accuracy, sparseFARE obtains fairer results (see Appendix D).

**Efficiency Analysis.** Our attention-based methods are more computationally efficient than the kernel-based baselines. CCLK, by requiring matrix inversion, costs $O(b^3)$, while FARE costs $O(b^2)$ and SparseFARE costs $O(b \log(b))$ (see Appendix B).

## 5    RELATED WORK

The majority of the literature on fair contrastive learning has considered only binary protected attributes (Park et al. (2022); Chai & Wang (2022)). With binary protected attributes, debiasing can be achieved by forming positive pairs as samples with opposing bias classes (Cheng et al. (2021); Hong & Yang (2021); Shen et al. (2021)). Another approach is to form positive pairs by using auxiliary models to learn optimal augmentations that obfuscate the bias class of the sample (Ling et al. (2022); Zhang et al. (2022)). This paper proposes an attention-based framework to deal with more general notions of fairness that accommodate high cardinality or continuous protected attributes, whereby we learn semantically meaningful representations such that the protected information has been removed. Tsai et al. (2022) also consider this setting and use kernel similarity functions to weigh negative samples along the bias dimension for contrastive learning. Our approach differs from their method by using an attention mechanism to learn the bias-causing interactions among samples without specifying a pre-defined kernel.

Our paper also connects to the growing literature surrounding kernel and attention. Most existing work has looked at decomposing the attention computation and enriching or explaining this mechanism by interpreting it as a kernel function. Tsai et al. (2019) propose novel attention mechanisms based on differing kernel functions and Song et al. (2021) propose enriching attention with implicit kernel estimation, while Tao et al. (2023) explain attention through nonlinear SVD of asymmetric kernels and Wright & Gonzalez (2021) view attention as infinite-dimensional non-mercer binary kernel machines. In contrast, our work derives an attention mechanism from a kernel-based method to learn a task-specific similarity metric that can capture the bias-interaction structure and assist the training procedure to learn better-debiased representations.

Lastly, sparse attention has been studied in the context of efficient transformers. Sparsity in attention mechanisms has been implemented via sparse factorization (Child et al.), via local windows (Beltagy et al. (2020)), and via locality-sensitive hashing (Kitaev et al. (2020)). While our work leverages locality-sensitive hashing, it does not do so merely to save on computational costs. Rather, locality-sensitive hashing supplements the debiasing scheme by sparsifying the entries of the attention map corresponding to extreme bias-inducing samples. To the best of our knowledge, ours is among the early works of using locality-sensitive hashing, or sparsification in general, for learning fair representations.

## 6    CONCLUDING REMARKS

In this paper, we present the Fairness-Aware (FARE) attention mechanism, the Sparse Fairness-Aware (SparseFARE) attention mechanism, and the corresponding Fair Attention-Contrastive (Fare-Contrast) criterion for learning fair representations. We address the difficult problem setting of high cardinality or continuous protected attributes and show that FARE and SparseFARE are able to learn a similarity metric over protected attributes that captures the bias-causing interactions among samples, while also focusing on bias-causing samples that are confounding the model. As a result, our attention-based approach is able to learn debiased and semantically meaningful representations. A limitation of our method is that they only capture one attention pattern between protected attributes, thereby providing only one single context to condition the similarity scores. It is indeed necessary to extend FARE and SparseFARE to a multi-head attention setting to capture more diverse contextual representations. We leave this interesting research direction as future work.

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

# Appendix for "Fairness-Aware Attention for Contrastive Learning"

## Table of Contents

## A  EXPERIMENTAL DETAILS

This section provides the details of the model and training for experiments in Section 4.

### A.1  TRAINING AND EVALUATION

**ColorMNIST.** Samples in the colorMNIST dataset are 32x32 resolution handwritten digit images, where the digit is represented in black and the background is some known assigned color which is representable as a continuous RGB color vector. The train-test split is 60,000 training images to 10,000 test images. The augmentation scheme is randomized resized crop followed by a random horizontal flip. We pre-train using the LARS optimizer (You et al. (2017)) and cosine annealing for the learning rate scheduler. The full FARE attention mechanism with sparsification uses 8 rounds of hashing, a bucket size of 64, and backwards and forwards cross-bucket attention. The linear classifier is trained using L-BFGS as optimizer over 500 iterations. We pre-train with a batch size of 256 for 50 epochs.

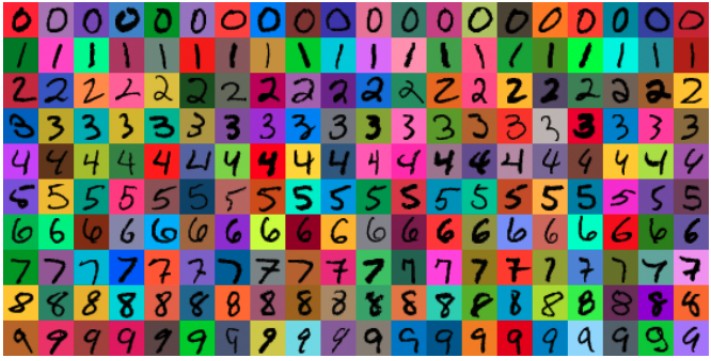

Figure 2: colorMNIST dataset (Tsai et al., 2022)

We follow the recent contrastive learning literature (Chen et al. (2020), Robinson et al. (2020), Wu et al. (2020)) and pre-train the full model before discarding everything except the backbone encoder at evaluation time.

**CelebA.** The train-test split is the default as provided by PyTorch. Images are resized to $128{\times}128$. Resnet-18 (He et al., 2016) is the encoder and we use the same 2-layer MLP and random augmentation strategies as Chen et al. (2020). Same as with colorMNIST, we pre-train with the LARS optimizer and use cosine annealing. We use a batch size of 512 and the LSH scheme uses buckets of size 128 with 8 rounds of hashing and backwards and forwards cross-bucket attention. We train the full model for 100 epochs and evaluate with a single linear layer trained on the frozen encodings for 10 epochs using Adam as optimizer.

To evaluate the fairness of the representations, we adopt the Equalized Odds (EO) metric (Hardt et al., 2016). Following Jung et al. (2022) and Zhang et al. (2022), we compute the metric over multiple sensitive attributes by:

$$\max_{\forall s^i, s^j \in S} \overline{\sum}_{\forall y, \hat{y}} \left| P_{s^i}\left(\hat{Y} = \hat{y}|Y = y\right) - P_{s^j}\left(\hat{Y} = \hat{y}|Y = y\right) \right|, \tag{14}$$

where $\overline{\sum}$ is the averaged sum, $Y$ is the target label, $\hat{Y}$ is the predicted label, and $s_i, s_j \in S$ are values of sensitive attributes. A smaller EO means a fairer model.

## A.2 Baselines

**ColorMNIST.** The relevant baselines for comparison are the InfoNCE model (**InfoNCE**) (Oord et al., 2018), the Fair-InfoNCE model with clustering (**Fair-InfoNCE**) (Tsai et al., 2022) and the conditional contrastive learning with kernel model (**CCLK**) (Tsai et al. (2022)).

The InfoNCE model uses the InfoNCE loss function 15 without performing any conditional sampling. The Fair-InfoNCE model uses the Fair-InfoNCE loss function 21 and performs conditional sampling by first clustering the protected attribute so as to discretize it and then sampling from within the same cluster as the anchor. We report this model's results according to its best performing cluster size as determined by its authors, which is found to be a 10-cluster partition. CCLK uses a kernel similarity metric for weighing negative samples in the batch according to their similarity in the bias-dimension. We report its results according to its best performing kernel choice as chosen by its authors which was the cosine kernel.

The InfoNCE objective (Oord et al., 2018) used in the baseline model InfoNCE is given by:

$$\sup_{f} \mathbb{E}_{(x,y_{pos})\sim P_{XY}, \{y_{neg}\}_{i=1}^n \sim P_Y^{\otimes n}} \left[ \log \frac{e^{f(x,y_{pos})}}{e^{f(x,y_{pos})} + \sum_{i=1}^{b} e^{f(x,y_{neg,i})}} \right] \tag{15}$$

**CelebA.** We compare with **SimCLR** (Chen et al., 2020) and all kernel implementations of CCLK provided by Tsai et al. (2022). For each kernel model, the kernel in the name refers to the what kernel similarity metric is chosen for measuring the similarity across protected attributes, which then determines the relevance of that sample for being contrasted with the positive sample. For example, CCLK-RBF uses the RBF kernel to compute similarity between two protected attributes.

# B CONNECTION BETWEEN KERNEL-BASED SCORING FUNCTION ESTIMATION IN (TSAI ET AL., 2022) AND ATTENTION

The CCLK model uses the following kernel-based scoring function estimation:

**Proposition 2** (Kernel-Based Scoring Function Estimation (Tsai et al., 2022)). *Given* $\{x_i, y_i, z_i\}_{i=1}^b \sim P_{XYZ}^b$, *the similarity score of the data pair* $(x_i, y)$ *given the anchor* $z_i$ *is computed via the finite-sample kernel estimation* $e^{f(x_i,y)}$ *when* $y \sim P_{Y|Z=z_i}$ *as follows:*

$$e^{f(x_i,y)} = \left[ K_{XY}(K_Z + \lambda \boldsymbol{I})^{-1} K_Z \right]_{ii}, \tag{16}$$

for $i = 1, \ldots, b$, $[K_{XY}]_{ij} := e^{f(x_i,y_j)}$, and $[K_Z]_{ij} := \langle \gamma(z_i), \gamma(z_j) \rangle_{\mathcal{G}}$, where $\gamma$ is some kernel feature embedding, $\mathcal{G}$ is the corresponding Reproducing Kernel Hilbert Space (RKHS), and $\langle \cdot, \cdot \rangle_{\mathcal{G}}$ is an inner product in space $\mathcal{G}$.

First, in comparison to Eqn. 2, FARE and sparseFARE avoid matrix inversion. FARE's attention computation has complexity $O(b^2)$ (Vaswani et al., 2017) and sparseFARE has complexity $O(b \ logb)$ (Kitaev et al., 2020), which improve significantly over $O(b^3)$ in Eqn. 2.

Second, our methods do not impose assumptions on the bias-causing interactions over protected attributes. In particular, we avoid specifying any particular kernel and allow our attention mechanism to learn the bias-causing interactions. To see this difference, we decompose the estimator in Eqn. 2 as follows:

$$\left[K_{XY}(K_Z + \lambda\mathbf{I})^{-1}K_Z\right]_{ii} \tag{17}$$
$$= [K_{XY}]_{i*}[(K_Z + \lambda\mathbf{I})^{-1}K_Z]_{*i}$$
$$= \sum_{j}^{b} w(z_i, z_j)e^{f(x_i, y_j)}, \tag{18}$$

where $w(z_i, z_j) = [(K_Z + \lambda I)^{-1}K_Z]_{ij}$ are smoothed kernel similarity scores (Tsai et al., 2022). Hence we see the (Tsai et al., 2022) estimator as performing a similar weighting of similarity scores between samples, with weights provided by the similarities over the protected attributes. This approach differs from ours however since the kernel must be pre-specified in $K_Z$. This imposes strong assumptions on bias-causing interactions that limit the extent to which the model can learn fair representations. Our method by contrast can be understood as replacing $w(z_i, z_j)$ with attention score $p(z_i, z_j)$. The attention mechanism can more flexibly model the bias-causing interactions and learns to focus-attention on bias-reducing samples that help learn the representation space.

We provide a proof adapted from (Tsai et al., 2022) of their kernel-based scoring function estimation below.

*Proof of kernel-based scoring function estimation.* First, letting $\Phi = [\phi(g(y_1)), \ldots \phi(g(y_b))]^\top$ be the matrix of kernel embeddings for encodings $g(y_i)$ with feature map $\phi$ and $\Gamma = [\gamma(z_1), \ldots, \gamma(z_b)]^\top$ be the matrix of kernel embeddings for protected attribute outcomes $z$ with feature map $\gamma$, Definition 3 provides the Kernel Conditional Embedding Operator (Song et al., 2013):

**Definition 3.** *[Kernel Conditional Embedding Operator (Song et al., 2013)] The finite-sample kernel estimation of $\mathbb{E}_{y \sim P_{Y|Z=z}}[\phi(g(y))]$ is $\Phi^\top(K_Z + \lambda I)^{-1}\Gamma\gamma(z)$ where $\lambda$ is a hyperparameter.*

Then, according to Definition 3, for any given $Z = z$, $\phi(g(y))$ when $y \sim P_{Y|Z=z}$ can be estimated by

$$\Phi^\top(K_Z + \lambda\mathbf{I})^{-1}\Gamma\gamma(z) \tag{19}$$

We look for the inner product between (5) and the encoding of $(x_i, z_i)$ when $y \sim P_{Y|Z=z_i}$:

$$\langle\phi(g(x_i)), \Phi^\top(K_Z + \lambda\mathbf{I})^{-1}\Gamma\gamma(z_i)\rangle_{\mathcal{H}} = tr\left(\phi(g(x_i))^\top \Phi^\top(K_Z + \lambda\mathbf{I})^{-1}\Gamma\gamma(z_i)\right)$$
$$= [K_{XY}]_{i*}(K_Z + \lambda\mathbf{I})^{-1}[K_Z]_{i*} = [K_{XY}]_{i*}[(K_Z + \lambda\mathbf{I})^{-1}K_Z]_{*i}$$
$$= [K_{XY}(K_Z + \lambda\mathbf{I})^{-1}K_Z]_{ii} \tag{20}$$
$$\square$$

## C  COMPARISON OF FAIR-INFONCE AND FARECONTRAST

We present a discussion of the differences between the Fair-InfoNCE objective from Tsai et al. (2021b) and the FAREContrast objective we use to train our attention-based FARE models. FARE-Contrast is derived from Fair-InfoNCE by replacing the conditionally sampled negative pairs with the output of the FARE attention mechanism. This leads to a difference firstly in sampling procedure and secondly in the inclusion of learnable attention scores in the loss.

The Fair-InfoNCE (Tsai et al., 2021b) is given as:

$$\sup_f \mathbb{E}_{z \sim P_Z, (x, y_{pos}) \sim P_{XY|Z=z}, \{y_{neg}\}_{i=1}^b \sim P_{Y|Z=z}^{\otimes b}} \left[\log \frac{e^{f(x, y_{pos})}}{e^{f(x, y_{pos})} + \sum_{i=1}^b e^{f(x, y_{neg,i})}}\right], \tag{21}$$

and FAREContrast is given as:

$$\sup_f \mathbb{E}_{\{(x_i,y_i,z_i)\}_{i=1}^b \sim P_{XYZ}^{\otimes b}} \left[ \log \frac{e^{f(x_i,y_i)}}{e^{f(x_i,y_i)} + \sum_{j=1}^b \text{softmax}\left((W_Q z_i)^\top W_K z_j / \rho\right) e^{f(x_i,y_j)}} \right], \quad (22)$$

where $b$ denotes the batch size, $f : \mathcal{X} \times \mathcal{Y} \to \mathbb{R}$ is a mapping given by $f(x,y) = $ cosine similarity$\left(g_{\theta_X}(x), g_{\theta_Y}(y)\right)/\tau$, $g_{\theta_X}, g_{\theta_Y}$ are neural networks parameterized by $\theta_X, \theta_Y$, and $\tau$ is a hyperparameter scaling the cosine similarity.

We see that FAREContrast does not require conditional sampling of the negatively paired samples, $\{y_{neg}\}_{i=1}^b \sim P_{Y|Z=z}^{\otimes b}$ for outcome of the of the protected attribute $z$. Instead, FARE considers the whole batch and selectively weights samples according to their protected attribute status. One issue with conditional sampling as in Eqn. 21 is data scarcity, whereby conditioning on $Z = z$ can lead to insufficient negative samples for contrasting (Tsai et al., 2022). This problem is exacerbated when the protected attribute has high cardinality or is continuous, which is the problem setting we aim to deal with. When there are insufficient negative samples, we incur risk of poorly learnt representations and collapse (Chen et al., 2020; Chen & He, 2021). For this reason, we derive FARE which considers the whole batch and uses learnt attention scores to accentuate/attenuate negative samples according to their bias characteristics.

The second difference is then the attention weights included in FAREContrast. Including the attention weights in FAREContrast means that that FARE learns according to information coming from the gradients and so can better focus on samples that help minimize the loss, thereby helping the encoder to learn meaningful representations.

# D ADDITIONAL RESULTS

## D.1 LSH BUCKET SCHEME

| Attention Scheme | Top-1 Test Accuracy (↑) | Bias Removal (↑) |
|---|---|---|
| Adjacent | $86.4 \pm 1.3$ | $74.0 \pm 3.8$ |
| Intra | $84.9 \pm 2.1$ | $58.2 \pm 9.8$ |

Table 3: Sparsification Scheme on ColorMNIST Results. Bias removal is measured by MSE, where high MSE indicates more color information has been removed from the learned representations.

Table 3 shows results for when the LSH scheme considers intra-bucket attention versus the standard adjacent bucket attention (where attention is computed across adjacent buckets). We see fairly substantial drop in performance when restricting attention to within the same bucket, both in terms of accuracy and fairness. Lower accuracy is intuitive given the intra-bucket attention removes three quarters of negative samples, which depletes the model's ability to learn meaningful representations. At the same time, we see lower fairness, despite the heavy debiasing scheme. This may support the conclusion that to learn effectively debiased representations, the model needs sufficiently many samples to learn to attend over and focus on bias-reducing samples. With too few samples in the batch, the model is ignoring too many samples, including ones that would help it learn debiased representations.

## D.2 FAIRNESS-ACCURACY TRADEOFF

The two metrics that capture both representation quality and fairness are Accuracy and Equalized Odds (EO). Table 2 showed that SparseFARE Pareto dominates all kernel baselines in terms of both fairness and accuracy, with the exception of CCLK-Linear and CCLK-Polynomial, which were able to attain slightly higher accuracy. We therefore further compare SparseFARE to these two models by plotting the fairness-accuracy tradeoff curves in Figure **??**. The curves are produced by plotting EO and Accuracy at four stages during training - after 25, 50, 75, and 100 epochs. We see that for every level of accuracy, SparseFARE achieves better fairness (lower EO). This implies that SparseFARE attains a better fairness-accuracy tradeoff. Additionally of interest, we find that

SparseFARE is even able to simultaneously minimize EO while increasing accuracy, implying that it can learn representations that do not necessarily need to compromise fairness for higher accuracy.

### D.3 COMPARISON WITH WORK IN PARTIAL ACCESS TO SENSITIVE ATTRIBUTES

| Model | Test Accuracy ($\uparrow$) | EO ($\downarrow$) |
|---|---|---|
| *Supervised Models* | | |
| CGL + G-DRO (Sagawa et al., 2019) | 71.4 | 21.9 |
| CGL + FSCL (Park et al., 2022) | 74.0 | 25.6 |
| *Unsupervised Models* | | |
| CGL + VFAE (Louizos et al., 2015) | 72.7 | 28.7 |
| CGL + GRL (Raff & Sylvester, 2018) | 73.8 | 26.9 |
| SimCLR (Chen et al., 2020) | **77.7** | 39.6 |
| FairCL (Zhang et al., 2022) | 74.1 | 24.5 |
| FARE (**ours**) | 73.7 | 23.5 |
| SparseFARE (**ours**) | 70.4 | **18.7** |

Table 4: CelebA Results. Fare and SparseFARE in comparison with unsupervised and supervised models under partial sensitive label access.

This paper uses the same experimental setup on CelebA as Zhang et al. (2022) in terms of training procedure and evaluation protocol. Zhang et al. (2022) differs, however, in the sense that the authors assumes only partial access to sensitive attributes and therefore use auxiliary models, for example an editor (Zhang et al., 2022) or CGL (Jung et al., 2022), to solve this problem. Given the experimental setups are the same, we include their results as well for reference, however we do not feature these results in the main body given the important difference regarding sensitive attribute access.

## E FAIR ATTENTION-CONTRASTIVE CRITERION

We do not include a learnable value transformation $W_V$ on the raw similarity scores such that $V = UW_V$ where $U = [e^{f(x_i,y_j)}]_{ij}$ as doing so allows the optimization process to obtain 0 loss without learning meaningful representations. This is seen immediately from the criterion, where allowing $W_V$ gives individual similarity scores as $w_{ij}e^{f(x_i,y_j)}$ in the criterion:

$$\sup_{f} \mathbb{E}_{\{(x_i,y_i,z_i)\}_{i=1}^{b} \sim P_{XYZ}^{b}} \left[ \log \frac{e^{f(x_i,y_i)}}{e^{f(x_i,y_i)} + \sum_{j \in S_i} p(z_i, z_j) w_{ij} e^{f(x_i,y_j)}} \right]$$

hence the loss is minimised by sending $w_{ij} \to \infty \ \forall i, j$.

# F   ETHICAL CONSIDERATIONS

We note that there are two, interconnected prevalent ethical issues in fair ML. The first is that almost all fair ML literature simplifies the problem of fairness to simple binaries and the second is that fairness metrics (which are typically built atop these binaries) and the choice of which to use themselves involve value judgements that can disadvantage certain people. People have intersectional identities and invariably belong to multiple groups simultaneously. When it comes to choosing fairness metrics, inherent to the majority of approaches in fair ML is that the researcher or practitioner decide what definition of fairness to use for their model. It has been shown that various definitions of fairness are not only mutually inconsistent but also prioritise different groups in different scenarios (Garg et al., 2020). In a sense then, solving for fairer ML models only pushes the problem from the model and onto the practitioner, as a 'fairer' model itself advantages and disadvantages different groups under different settings.

These two ethical considerations motivate the approach of our paper to conceptualise fairness in a more general setting where sensitive attributes can be continuous and multidimensional and fairer models are measured in terms of sensitive information removal. This conception avoids the ethical issues of binaries and fairness metrics.

We do note however that there still exist ethical concerns with our approach in terms of explainability. Measuring fairness by sensitive information removal (by measuring loss from a trained classifier) does not have an intuitive scale or unit of measurement for discussing the fairness or unfairness of a model. Although we can compare two models in terms of which is fairer, saying a model is fair because it scores some number in MSE has little intuitive meaning. Being unable to communicate the specifics of how a learned representation has removed sensitive information and how will affect downstream classifiers risks undermining confidence in fair ML as well.

Despite the explainability issue, we nonetheless believe that this approach represents a promising and exciting direction in fair ML that deal with substantive existing ethical issues. We hope that one area of future research may be deriving theoretical frameworks that can derive guarantees between sensitive information removal from debiased representations and upper bounds on downstream fairness metrics. This would develop a practical link to well-

known ideas of fairness and how unfair out-
comes could appear in worst-case scenarios.

