# OpenReview forum: "Fairness-Aware Attention for Contrastive Learning"
_ICLR.cc/2024/Conference — Submitted to ICLR 2024_

### Official Review · Reviewer_K853 · 2023-10-28

**Soundness:** 2 fair
**Presentation:** 2 fair
**Contribution:** 2 fair
**Rating:** 5
**Confidence:** 2

**Summary:**

This paper explores the combination of fairness methods and attention-based techniques in machine learning to reduce bias and improve model effectiveness. It proposes innovative approaches to minimize bias in machine learning algorithms, particularly in the context of graph-based data. The paper employs attention mechanisms to guide the model to focus on data that is less likely to introduce bias. It also discusses the role of contrastive learning in bringing similar data points closer together in the feature space, contributing to a more equitable model. The paper provides a thorough technical foundation, making it a valuable guide for those interested in the topic.

**Strengths:**

The paper is notable for its creative fusion of methods to improve fairness and attention-based techniques to combat bias, bringing a fresh perspective to existing research. It lays a solid technical foundation, serving as a detailed guide for those new to the field as well as seasoned experts. Given the growing emphasis on fairness in machine learning, the relevance of the paper is heightened.

**Weaknesses:**

The paper falls short in clearly describing the empirical tests conducted to validate its findings, leaving room for improvement. Questions about the scalability of the proposed methods also remain unanswered, making it uncertain how they would perform on larger datasets or in different domains. In addition, the paper doesn't address the potential trade-offs between fairness and other issues such as accuracy, nor does it explore the ethical considerations associated with using machine learning to reduce bias.

**Questions:**

Please see the Strengths and Weaknesses sections.

---

> ### Author Response · Authors · 2023-11-22
> **Response to Comments**
>
> **Need for further empirical results and experimental details**
>
> Please see the general comment for all reviewers addressing these points.
>
> **What are the potential trade-offs between and fairness and accuracy?**
>
> **Answer**
>
> We thank the reviewer for drawing our attention to fairness-accuracy tradeoff. For the colorMNIST task, we find that sparseFARE obtains top fairness and (shared) top accuracy. This implies sparseFARE weakly Pareto dominates the other baseline models InfoNCE, Fair-InfoNCE, and CCLK. For the CelebA task, we observe that SparseFARE Pareto dominates all the kernel models except for Linear and Polynomial. SparseFARE nonetheless attains a better fairness-accuracy tradeoff than Linear and Polynomial and so is still preferable. Additionally noteworthy is that over the course of training we observe sparseFARE improving both accuracy and fairness together, implying that sparseFARE is able to learn representations that do not necessarily need to compromise fairness for higher accuracy.
>
>
> **What are the ethical considerations of using ML to reduce bias?**
> **Answer**
>
> We thank the reviewer for pointing out this important consideration. We have included the following discussion in Appendix E. We present the discussion here as well for convenience.
>
> In terms of ethical considerations of using machine learning for bias reduction, we argue that fairness-aware machine learning is a powerful and ethically well-motivated approach to the problem of unfair outcomes in AI, and in particular the conceptualization of the problem we use and our proposed methods also offer benefits in terms of ethics.
>
> We note that there are two, interconnected prevalent ethical issues in fair ML. The first is that almost all fair ML literature simplifies the problem of fairness to simple binaries and the second is that fairness metrics (which are typically built atop these binaries) and the choice of which to use themselves involve value judgements that can disadvantage certain people. People have intersectional identities and invariably belong to multiple groups simultaneously. When it comes to choosing fairness metrics,  inherent to the majority of approaches in fair ML is that the researcher or practitioner decide what definition of fairness to use for their model. It has been shown that various definitions of fairness are not only mutually inconsistent but also prioritise different groups in different scenarios [1]. In a sense then, solving for fairer ML models only pushes the problem from the model and onto the practitioner, as a ‘fairer’ model itself advantages and disadvantages different groups under different settings.
>
> These two ethical considerations motivate the approach of our paper to conceptualise fairness in a more general setting where sensitive attributes can be continuous and multi-dimensional and fairer models are measured in terms of sensitive information removal. This conception avoids the ethical issues of binaries and fairness metrics.
>
> We do note however that there still exist ethical concerns with our approach in terms of explainability. Measuring fairness by sensitive information removal (by measuring loss from a trained classifier) does not have an intuitive scale or unit of measurement for discussing the fairness or unfairness of a model. Although we can compare two models in terms of which is fairer, saying a model is fair because it scores some number in MSE has little intuitive meaning. Being unable to communicate the specifics of how a learned representation has removed sensitive information and how will affect downstream classifiers risks undermining confidence in fair ML as well and creating accountability problems as well.
>
> Despite the explainability issue, we nonetheless believe that this approach represents a promising and exciting direction in fair ML that deal with substantive existing ethical issues. We hope that one area of future research may be deriving theoretical frameworks that can derive guarantees between sensitive information removal from debiased representations and upper bounds on downstream fairness metrics. This would develop a practical link to well-known ideas of fairness and how unfair outcomes could appear in worst-case scenarios.
>
> [1]: Garg, Pratyush, John Villasenor, and Virginia Foggo. "Fairness metrics: A comparative analysis." 2020 IEEE International Conference on Big Data (Big Data). IEEE, 2020.

---

### Official Review · Reviewer_Nbdh · 2023-11-01

**Soundness:** 1 poor
**Presentation:** 1 poor
**Contribution:** 2 fair
**Rating:** 3
**Confidence:** 4

**Summary:**

This work is concerned with the problem of fair representation learning, and in particular with how to debias representations in contrastive self-supervised learning. The authors identify limitation with current approaches, in that modelling assumptions about bias attribute are too strong, and they suggest instead a way to condition similarity scores between pairs of positives (or negatives) to a bias attribute via a proposed variant of a “self-attention” mechanism. At the same time, they extend their architectural intervention to a sparsified attention scheme using locality-sensitive hashing which the goal of masking out interactions between pairs which may help with the task. In addition, they propose alternative contrastive learning losses for training with supervision from the bias attribute.

**Strengths:**

Creating methods for debiasing representations learnt from training datasets containing spurious-correlations, label-imbalances, or sensitive attributes is an important problem.

The authors use existing literature on the relation between self-attention operator and kernels [1] to derive a similarity score for pairs which is conditioned on bias attribute information. Exploring new formulations of conditional similarity scores can be an interesting avenue.

[1] Yao-Hung Hubert Tsai, Shaojie Bai, Makoto Yamada, Louis-Philippe Morency, and Ruslan Salakhutdinov. 2019. Transformer Dissection: An Unified Understanding for Transformer’s Attention via the Lens of Kernel. In Proceedings of the 2019 Conference on Empirical Methods in Natural Language Processing and the 9th International Joint Conference on Natural Language Processing (EMNLP-IJCNLP).

**Weaknesses:**

1. The paper is poorly written. There are notation problems (e.g. in section 2 the matrix $U$ of similarity scores is overloaded, in section 3.3 there is no $z_i$ appearing in the loss even if it is sampled), incorrect math statements (like in section 3.1 that $\phi(g(y))$ is used to estimate $\mathbb{E}_{y|z} \phi(g(y))$), missing important citations (like [1] for the derivations in page 5 of attention as a kernel-based similarity), and often times definitions (such as for the dataset used and the “bias removal” evaluation metric) are not self-contained in the paper.
2. Novelty concerns: the FAREContrast objective function is essentially the same as the one described at [2].
3. Sparsifying the attention is poorly motivated, and it incurs a considerable implementation cost for the induced performance benefit over the considered baselines.
4. Empirical evaluation is very limited. The authors consider a variant of ColorMNIST, which is not described in the paper, and measure the top-1 test accuracy and a bias removal evaluation metric, which is not described. The paper needs to consider more benchmarks, such as CelebA (classifying hair color while the sensitive attribute is gender) [see benchmarks, 3], and evaluate according to fair/group-robust classification performance metrics (instead of iid accuracy), such as a group-balanced (or worst-case) test accuracy (depending on the dataset) and/or Equalized Odds [4, see 5 on how it is applied].

[2] Yao-Hung Hubert Tsai, Martin Q Ma, Han Zhao, Kun Zhang, Louis-Philippe Morency, and Ruslan Salakhutdinov. Conditional contrastive learning: Removing undesirable information in self- supervised representations. arXiv preprint arXiv:2106.02866, 2021c.
[3] Sagawa, Shiori, et al. "Distributionally robust neural networks for group shifts: On the importance of regularization for worst-case generalization." arXiv preprint arXiv:1911.08731 (2019).
[4] Hardt, Moritz, Eric Price, and Nati Srebro. "Equality of opportunity in supervised learning." Advances in neural information processing systems 29 (2016).
[5] Zhang, Fengda, et al. "Fairness-aware contrastive learning with partially annotated sensitive attributes." The Eleventh International Conference on Learning Representations. 2022.
[6] Tsai, Yao-Hung Hubert, et al. "Conditional contrastive learning with kernel." arXiv preprint arXiv:2202.05458 (2022)

**Questions:**

See Weaknesses above.

---

> ### Author Response · Authors · 2023-11-22
> **Response to Comments on Notation Problems**
>
> **Notation in section 2 for matrix U is overloaded**
>
> **Answer**
>
> We thank the reviewer for drawing our attention to this potential source of confusion. We noticed there is a potentially connected typo in which we wrote $U = [e^{f(x_i, y_j)}]_{ij} $ . As U is a matrix with $(i,j)$ element given by these similarity scores, this should of course be that the $[U]_ij = e^{f(x_i, y_j)}$. Regarding overloaded notation, the two references to the definition of U is simply first a general representation of U followed by an instantiation of what U is for our purposes. Specifically, the matrix U is introduced as the data matrix such that when transformed by the values projection matrix W_V, we obtain the values V. That is, $V = UW_V$. Given our FARE set up, the untransformed values in our case is the matrix of similarity scores. Hence in our model U is the matrix whose $(i,j)$ element is given by $e^{f(x_i, y_j)}$. In other words, naming U as the matrix of similarity scores $e^{f(x_i, y_j)} $ is just a specification of the data that takes the place of the values for our model.
>
>
> **In section 3.3, there is no $z_i$ appearing in the loss**
>
> **Answer**
>
> We thank the reviewer for pointing this out and have fixed the notation. In particular FARE is now defined as an operator that takes as argument the whole batch, $FARE( \{ (x_i, y_i, z_i) \}^b_{i=1} )$. This clarifies the fact that FARE computes a finite sample estimate of the conditional similarity score between $(x_i, z_i)$ and $y$ for $y \sim P_{Z = z_i}$ by taking in information over the whole batch and weighing each sample according to their protected attribute.
>
> **There are missing citations, for example [1], on page 5 for the derivation of attention as kernel-based similarity, for example**
>
> **Answer**
>
> We agree that [1] is highly relevant related work and indeed we include reference to it in our Related Work section. However, [1] is not cited here because our derivation doesn’t rely on theorems or results from [1] or other work, aside from the use of the kernel density estimator approach to estimate the joint and marginal densities which is cited. We also note that [1] is different to our work in the sense that [1] takes the attention mechanism and decomposes it into parts that, by replacing with various kernels, derive new and enriched attention mechanisms. Our work, on the other hand, doesn’t involve decomposing attention nor replacing any component of attention with kernel similarity metrics. Our work shows instead that the estimation of the similarity score (when conditioning on some additional attribute) in the contrastive objective can be approximated by an attention mechanism. In this sense we don’t perform any decompositions and we go straight from conceptualising the similarity score as an inner product to deriving an attention mechanism without appealing to kernel similarity metrics in between. For these reasons we agree that [1] is a relevant work to understand some of the research community’s work in enriching attention, however we do not cite it on this page due to the differences mentioned.
>
> **There is an incorrect math statement in section 3.1 that $\phi(g(y))$ is used to estimate $E_{y | z} [\phi(g(y)))] $**
>
> **Answer**
>
> We apologise for the confusion regarding notation. To clarify, we don’t use $\phi(g(y))$ to estimate $E_{y | z} [\phi(g(y)))] $. Rather, we use $E_{y | z} [\phi(g(y)))] $  to estimate $\phi(g(y))$. $\phi(g(y))$ is the unknown quantity of interest and we use its expectation to estimate it, expanding the definition of the expectation using joint and marginal densities which we then estimate with kernel density estimators. This allows us to derive an estimator of $\phi(g(y))$.
>
> [1]: Yao-Hung Hubert Tsai, Shaojie Bai, Makoto Yamada, Louis-Philippe Morency, and Ruslan Salakhutdinov. 2019. Transformer Dissection: An Unified Understanding for Transformer’s Attention via the Lens of Kernel. In Proceedings of the 2019 Conference on Empirical Methods in Natural Language Processing and the 9th International Joint Conference on Natural Language Processing (EMNLP-IJCNLP).

---

> > ### Comment · Reviewer_Nbdh · 2023-11-22
> > **Response to rebuttal (1/2)**
> >
> > **Notation in section 2 for matrix U is overloaded**
> >
> > The authors claim that $U$ is just a data specification of an abstract $U$. However, while it is explicitly mentioned in the paper that $U \in \mathbb{R}^{n \times d_m}$, in its specified form it is a $n \times n$ matrix. Furthermore, in the actual FAIR a $W_V$ matrix is not used as it leads to collapse. If anything, it seems that the actualized $U$ serves directly as $V$. As a result, the connection to the self-attention formulation is only loose and serves as an inspiration for FARE formulation. Many of these details need to be clarified further, and this part needs heavy rewriting for that.
> >
> > **There are missing citations, for example [1], on page 5 for the derivation of attention as kernel-based similarity, for example**
> >
> > To the reviewer’s perspective, [1] is exactly the inspiration for FARE here, as [1] provides with the way to explain attention as a kernelized non-linear similarity score, which is exactly what the authors deploy here. In other words, “the use of the kernel density estimator approach to estimate the joint and marginal densities” is central to both cases.
> >
> > **There is an incorrect math statement in section 3.1 that $\phi(g(y))$  is used to estimate $\mathbb{E} \phi(g(y))$**
> >
> > What is meant here in the original review is that **one can only say that $\phi(g(y))$  can be used to estimate $\mathbb{E} \phi(g(y))$**. The opposite as the authors claim in the paper does not make mathematical/statistical sense. One quantity is stochastic, $\phi(g(y))$, and the other is not, $\mathbb{E} \phi(g(y))$. One can use (via Monte Carlo)  $\phi(g(y))$ to stochstically approximate $\mathbb{E} \phi(g(y))$, the other way around does not make sense since  $\phi(g(y))$ is not even a constant. The justification here needs to be reconsidered.

---

> > > ### Comment · Reviewer_Nbdh · 2023-11-22
> > > **Response to rebuttal (2/2)**
> > >
> > > **What are the differences between FAREContrast and the objective in [2]**
> > >
> > > Please provide in exact latex the two losses and elucidate on the mathematical differences between the two formulations of the loss. As of now, when we abstract away the particularities of instantiating a similarity score (which we may nonetheless allow to depend of $z$), I am not able to spot any differences between the two.
> > >
> > > **Is sparseFARE sufficiently well motivated and does its induced performance benefit justify the implementation cost?**
> > >
> > > According to the results provided in the rebuttal, it is not true that “for CelebA, it Pareto dominates or exhibits a more favourable fairness-accuracy tradeoff than the baselines”, as the SimCLR baseline outperforms all methods in the iid accuracy. The claim needs to be softened and furthermore the reviewer believes that further empirical evidence needs to be provided for a paper which focuses on fairness of representations.
> > >
> > > In addition, the bias removal metric needs also to be clarified and described in the paper.
> > >
> > > Overall, for the reasons explained above, the reviewer chooses to maintain their original assessment.
> > >
> > > The paper can improve significantly with better and correct writing describing the method, better attribution to relevant literature, and more empirical evidence justifying the additional incurring cost of SparseFARE.

---

> > > > ### Author Response · Authors · 2023-11-23
> > > > **Response to comment 2/2**
> > > >
> > > > **What are the differences between FAREContrast and the objective in [2]**
> > > >
> > > > **Answer**
> > > >
> > > > As suggested, we now include in Appendix C a side-by-side comparison of Fair-InfoNCE [2] and FAREContrast. In short, aside from the inclusion of the learnable attention scores, the main difference comes from the replacement of the conditional sampling procedure of [2]. Furthermore, we acknowledge of course that FAREContrast is derived from Fair-InfoNCE, and so can be understood as a variant of it, whereby using attention to avoid conditional sampling brings about our objective.
> > > >
> > > > **Is sparseFARE sufficiently well motivated and does its induced performance benefit justify the implementation cost?**
> > > >
> > > > **Answer**
> > > >
> > > > We agree with the reviewer that the correct claim is that our results show SparseFARE improves only over the kernel baselines, and not the SimCLR baseline due to its high accuracy. Indeed, this was the claim in our results section in the main body - the false claim of improving over all baselines was just a misphrasing in the previous response here, we apologize for that.

---

> > > ### Author Response · Authors · 2023-11-23
> > > **Response to comment 1/2**
> > >
> > > **Notation in section 2 for matrix U is overloaded**
> > >
> > > **Answer**
> > >
> > > We thank the reviewer for providing additional information on this point. We see the need for clarification as suggested by the reviewer and have rewritten the attention background section to clarify $U$ and its definition, in particular emphasising up front the departures from typical self-attention.
> > >
> > > **There are missing citations, for example [1], on page 5 for the derivation of attention as kernel-based similarity**
> > >
> > > **Answer**
> > >
> > > We thank the reviewer again here for the additional information regarding the similarities between our work and [1]. We have duly updated this section to state up front before we begin the derivation that our work leverages attention as kernelized non-linear similarity scores with the appropriate citation.

---

> ### Author Response · Authors · 2023-11-22
> **Response to General Comments**
>
> **Need for further empirical results and experimental details**
>
> Please see the general comment for all reviewers addressing these points.
>
> **What are the differences between FAREContrast and the objective in [2]**
>
> **Answer**
>
> The key difference between the objective in [2] and FAREContrast is that FAREContrast contains learnable attention scores as conditioning coefficients on the sample similarity scores whereas [2] does not. [2] therefore is used only to train in the MLP and encoder where FAREContrast trains the MLP, encoder, and attention projection matrices in one end-to-end loop.
>
> **Is sparseFARE sufficiently well motivated and does its induced performance benefit justify the implementation cost?**
>
> **Answer**
>
> We argue that the benefits of sparseFARE do indeed justify its implementation cost. In particular, we note that in colorMNIST sparseFARE Pareto dominates all models and is significantly more computationally more efficient than the kernel models. For CelebA, it Pareto dominates or exhibits a more favourable fairness-accuracy tradeoff than the baselines. We argue that this is because sparisification makes intuitive sense within this framework, whereby samples that are sufficiently distant in terms of bias-status (and therefore likely to bias the representations) may as well be dropped and allow attention to be focussed over the remaining samples that help learn debiased and meaningful representations.
>
> Secondly, attention sparsification schemes typically exhibit a tradeoff between computational complexity and model performance (in this case taking both accuracy and fairness together) as sparisifation sacrifices the capacity of the attention mechanism for speed. SparseFARE, however, exhibits higher performance than the baselines while being faster as well and so again doesn’t exhibit the normal tradeoff we observe in the literature.
>
> Hence we believe that since sparsity, in this setting, is able to offer improvements in all these areas simultaneously, it’s implementation cost is justified.
>
> [2] Yao-Hung Hubert Tsai, Martin Q Ma, Han Zhao, Kun Zhang, Louis-Philippe Morency, and Ruslan Salakhutdinov. Conditional contrastive learning: Removing undesirable information in self- supervised representations. arXiv preprint arXiv:2106.02866, 2021c.

---

### Official Review · Reviewer_nzY3 · 2023-11-07

**Soundness:** 3 good
**Presentation:** 1 poor
**Contribution:** 2 fair
**Rating:** 3
**Confidence:** 3

**Summary:**

This paper aims to learn fair feature representations through contrastive learning.
Similar to [1], the authors adopt a learning scheme that assigns weights to data pairs according to the similarity of sensitive attributes. This is based on the assumption that the samples with similar sensitive attributes will serve as 'bias-reducing samples', which is beneficial for learning fair representations. The proposed method, FARE, for estimating the (conditional) similarity between the anchor and the negative samples utilizes attention-based weights instead of kernel-based weights [1]. The authors also propose an additional method, SparseFARE, that further sparsifies the attention map by discarding ‘extreme bias-causing’ samples. However, the experiments section seems incomplete, as the comparison with baselines is carried out solely on a synthetic dataset, and some important details about the experimental set-up are not provided.

**Strengths:**

* While existing works in fair contrastive learning often assume binary sensitive attribute setting, the two proposed approaches can be applied to settings with high-dimensional and continuous sensitive attributes.
* When the only available data is the batch of triplets $\set{(x_{i}, y_{i}, z_{i})}_{i=1}^{b}$, the conditional sampling procedure in the Fair-InfoNCE objective [2] can be addressed through the proposed attention-based approaches.
* The kernel-based method assumes a pre-defined kernel for calculating similarity, but attention-based methods learn similarity adaptively from the task, which alleviates the need for such an assumption.
* Attention-based methods can lead to improved computational complexity ($O(b^2)$ or $O(b\log{b})$) compared to the kernel-based methods ($O(b^3)$).

**Weaknesses:**

* The comparison with baselines is carried out solely on a synthetic dataset, ColorMnist [1]. Since this work is about learning fair representations, it seems necessary to consider experiments on fairness datasets (e.g., COMPAS, Adult), which are commonly used in the fairness literature, and to employ fairness criteria (e.g., Demographic Parity, Equalized Odds) for comprehensive assessment. Plotting a Pareto-frontier curve is an effective way to compare, especially when considering the accuracy-fairness trade-off.
* Some important details for the proposed method such as the model architecture, batch size, and hyperparameter selection are not provided. For clarity and to ensure the paper is self-contained, it would be better to describe the specific procedures used.
* Table 1 shows the result for CCLK [1] when using Cosine kernel, but [1] also provides a result for CCLK when Laplacian kernel is applied, showing Top-1 Accuracy of $85.0 \pm 0.9$ and MSE of $72.8 \pm 13.2$. Then, I'm not sure whether FARE indeed alleviates a significantly larger amount of bias compared to the baseline methods.
* Given that the performance gain doesn’t seem to be significant, it is not yet clear to me the benefits of the attention-based approach compared to the kernel-based approach. The kernel-based method relies on choosing an appropriate kernel, whereas the attention-based method focuses on training the model using data. However, it seems that more justification is required for the proposed method. It would be beneficial to include additional intuitive explanations on why attention-based methods are more effective than kernel-based methods for calculating the similarity of sensitive attributes, along with experimental results to support this. For instance, in Adult dataset, if ‘age’ is selected as the sensitive attribute, one could consider showing experimentally that the attention score tends to be higher when two individuals have similar ages, whereas this may not be the case with kernel-based methods.
* Minor suggestions
    * (p.4) “~ Fair-InfoNCE objective in Eqn. 1” → “~ Fair-InfoNCE objective in Eqn. 1.”
    * (p.5) “Given 6 and the kernel density estimators in 7,” → “Given Eqn. 6 and the kernel density estimators in Eqn. 7,”
    * (p.8) “Hence we only consider need to consider ~” → “Hence we only consider ~”
    * (p.14) Consider adding 'Eqn.' for consistency.

**Questions:**

* In Table 1, the result for SpareFARE appears to use the adjacent bucket scheme, but it differs from the result in Table 2 of the appendix. Which is correct?
* In Eqn. (12), does FARE use a feature map associated with the Cosine kernel for $\phi$?


[1]: Tsai et al. Conditional Contrastive Learning with Kernel. ICLR, 2022.

[2]: Tsai et al. Conditional Contrastive Learning for Improving Fairness in Self-Supervised Learning. Arxiv, 2021.

---

> ### Author Response · Authors · 2023-11-22
> **Response to comments**
>
> **Need for further empirical results and experimental details**
>
> Please see the general comment for all reviewers addressing these points.
>
> **Considering the performance of the Laplacian kernel, does FARE alleviate significantly more bias? If the performance gain isn’t significant, what are the benefits of the attention-based approach compared to the kernel-based approach?**
>
> **Answer**
> It’s true that when implementing CCLK with a Laplacian kernel, the baseline model achieves bias removal close to SparseFARE. We note that our method nonetheless provides 3 simultaneous benefits over the Laplace method in particular, and over the baselines more generally:
> Using the Laplacian method can indeed remove similar bias to our proposed models, but it does so at the expense of accuracy, shown by the drop in accuracy of 1.4%. Our SparseFARE does not suffer from this tradeoff, producing the same high bias removal with high accuracy concurrently. In this sense, we find that sparseFARE still Pareto dominates CCLK and the other baseline models.
> An additional benefit of our approach is ease of implementation. In order to discover the best performing kernel baseline method, the user would have to exhaustively search over candidate kernels, each with their own hyperparameters, where there is in general little intuitive guidance in terms of knowing prior to the problem which kernel is suitable. Our approach does not need to consider a host of similarity metrics since the attention mechanism can adaptively learn it given the task, and so in this sense our model is easier to implement and fine-tune.
> Lastly, we also refer to the benefits in computational complexity of our approach in reducing cubic computational complexity to $O(b^2)$ or $O(b log b)$.
>
> Overall, we agree that there probably will exist situations in which a pre-defined kernel performs close to our proposed method in terms of debiasing. In such situations we note that our methods can and do indeed match that debiasing performance without sacrificing accuracy, are faster to implement and fine-tune, and less computationally complex.
>
> **What are additional intuitive explanations for why attention is more effective than kernel for calculating similarity of sensitive attributes?**
>
> **Answer**
>
> An alternate way of seeing the benefit of attention is that it is trained in an end-to-end manner alongside the encoder whereas any given kernel is simply pre-specified. In this way, the attention mechanism helps to focus attention on samples that learn meaningful representations. For example, suppose in a contrastive step that in a given batch there are n samples with highly similar sensitive attributes to the anchor. A kernel would indiscriminately assign all these n samples high weights irrespective of those samples’ effect on the loss. Attention, by taking in information from the loss, is adapted to not only measure similarity over protected attributes, but be adapted by information coming from the gradients to better focus on samples that help minimize the loss and thereby learn meaningful representations.
>
> **In Table 1, the result for SpareFARE appears to use the adjacent bucket scheme, but it differs from the result in Table 2 of the appendix. Which is correct?**
>
> **Answer**
>
> We thank the reviewer for picking this up and we apologize for the oversight. The original number in the main body is correct - we’ve made the correction.
>
> **In Eqn. (12), does FARE use a feature map associated with the Cosine kernel for $\phi$ ?**
>
> **Answer**
>
> $\phi$ is the feature map associated with the exponentiation of the cosine kernel, which is what we need to aim for to estimate the similarity score in the objective. You can refer to equation 3 on page 3 to see the definition.

---

> ### Comment · Reviewer_nzY3 · 2023-11-22
> **Reply for author response**
>
> Thank you for the replies. I have some comments on Figure 3 and the new results on CelebA dataset.
>
> **Regarding Figure 3 in Appendix C.2:**
>
> * The blue curve in Figure 3 does not show an appropriate trade-off between fairness and accuracy, but rather shows an improvement in accuracy as fairness improves. Typically, accuracy decreases as fairness improves (e.g., Figure 6 of [3]).
> * This might be because the figure was plotted with a model that hadn't been fully trained yet.
> Namely, the authors chose to plot using a single model during its training, interrupted at 25, 50, 75, and 100 epochs.
> * However, I think that it makes sense to use different models, each with distinct hyperparameters, after they have reached convergence.
> Then, the author’s comments based on Figure 3 are not convincing.
>
> **On the new results with the CelebA dataset:**
> * In the additional experimental results on CelebA dataset, the authors used ‘Attractive' as the target attribute; however, since ‘Attractive’ is subjective and controversial, it is recommended to use one of the other 40 attributes in the dataset for discussion of fairness.
> * For instance, a setting that classifies hair color while incorporating gender as a sensitive attribute can be considered, as reviewer Nbdh suggested.
>
> [3]: Zhang et al. Fairness-aware Contrastive Learning with Partially Annotated Sensitive Attributes. ICLR, 2023.

---

> > ### Author Response · Authors · 2023-11-23
> > **Regarding the fairness-accuracy tradeoff curves**
> >
> > We thank the reviewer for the additional comments. We agree that considering converged models under different hyperparameter settings would indeed be informative for analysing fairness-accuracy tradeoffs. We do, nonetheless, posit that the graph still shows relevant information regarding the performance of SparseFARE and how it’s able surpass the kernel-baselines’ performance.
> >
> > First, regarding the slope of the tradeoff curve, we agree that typical fairness-accuracy tradeoff curves indicate that higher accuracy comes at the cost of lower fairness, however the definition of EO does allow for ‘backward’ sloping fairness-accuracy tradeoff curves. This is because EO is defined as the difference between true positive rates and false positive rates between non-protected and protected groups. This allows for more accurate models to also be fairer models when a model improves its accuracy on the protected groups. Indeed, we argue that this is what we observe by the slope of SparseFARE in the fairness-accuracy tradeoff graph. Hence the graph is intended to show that throughout training, the SparseFARE mechanism is successfully optimising more accurate and fairer representations, whereas the baselines do not.
> >
> > In this sense, we believe that considering the in-training performance is informative and that the slope of the tradeoff curve offers evidence towards the debiasing mechanism of SparseFARE, whereby it is improving predictive performance on the protected groups.

---

### Author Response · Authors · 2023-11-22
**Additional Experimental Results as Requested by All 3 Reviewers**

We thank all three reviewers for pointing out the need to reference some of the existing, more common fairness benchmarks and metrics  and to consider tradeoff curves. Our initial intention was to design a method that focussed on the problem setting of high cardinality and continuous protected attributes, which is why we originally omitted experiments on the more common benchmarks which use binary sensitive attributes. Nonetheless, we agree that this work is incomplete without checking its adaptability to these common benchmarks, especially those benchmarks outside the original intended scope of the model. To this end, we’ve included additional experimental results on the common benchmark CelebA, taking as target Attractive and combining Young and Male into sensitive attributes. We take Equalized Odds as the fairness metric. We chose this particular configuration of CelebA as it was the setup in [1], as recommended.

| **Model**               | **Acc. ($\uparrow$)** | **EO ($\downarrow$)** |
|-------------------------|-----------------------|------------------------|
| SimCLR [2] | **77.7** | 39.6 |
|-------------------------|-----------------------|------------------------|
|                         |                       |                        |
| *Kernel-based Models* [3] |                       |                        |
|-------------------------|-----------------------|------------------------|
| CCLK-Cosine              | 70.2                  | 22.4                   |
| CCLK-RBF                 | 69.9                  | 21.8                   |
| CCLK-Linear              | 71.1                  | 21.1                   |
| CCLK-Polynomial          | 71.0                  | 20.8                   |
| CCLK-Laplacian           | 70.0                  | 20.8                   |
|-------------------------|-----------------------|------------------------|
|                         |                       |                        |
| *Attention-based Models* |                       |                        |
|-------------------------|-----------------------|------------------------|
| FARE (**ours**)     | 73.7                  | 23.5                   |
| SparseFARE (**ours**)| 70.4                  | **18.7**               |


As our main contribution is to show that kernel-based conditional contrastive learning can be improved by using an attention mechanism instead, we take as baselines SimCLR and continue with the kernel baselines as proposed by [3]. Firstly and intuitively, SimCLR archives top accuracy and bottom fairness. Given Young and Male are highly correlated with the target, models that attempt to remove information regarding this sensitive attributes should be expected to suffer in terms of accuracy. Comparing over the kernel-based methods and attention-based methods, we see SparseFARE Pareto dominates all kernel methods except for Linear and Polynomial, which attain slightly higher accuracy. We therefore plot the fairness-accuracy tradeoff curves to show that SparseFARE is better in terms of tradeoff. For any level of accuracy, sparseFARE produces fairer outcomes. Additionally noteworthy is that over the course of training we observe sparseFARE improving both accuracy and fairness together, implying that sparseFARE is able to learn representations that do not necessarily need to compromise fairness for higher accuracy. Please see the updated paper and the additional results in the appendix for the graph.

We conclude from these results that again the attention-based methods offer strong improvements over kernel–based methods. We find evidence of fairer and more semantically meaningful representations, and at much lower computational complexity.

Additionally, given the experimental setup was adopted from [1] as requested, we include in Appendix C our results compared to the results in [1] for reference. We leave these results in the appendix as opposed to the main body since there is one key difference in that [1] assumes only partial access to sensitive attributes and so uses auxiliary models to deal with this problem setting.

[1]: Zhang, Fengda, et al. "Fairness-aware contrastive learning with partially annotated sensitive attributes." The Eleventh International Conference on Learning Representations. 2022.

[2]: Chen, Ting, et al. "A simple framework for contrastive learning of visual representations." International conference on machine learning. PMLR, 2020.

[3]: Tsai et al. Conditional Contrastive Learning with Kernel. ICLR, 2022.

---

### Author Response · Authors · 2023-11-22
**Omission of Key Details of Experimental Setup Now Corrected**

We thank all three reviewers for noting that the experimental section did not read self-contained. We have updated the experiments section with both the new results on CelebA and all relevant experimental details regarding datasets and evaluation protocol and metrics.

---

### Meta-Review · Area_Chair_Jdk4 · 2023-12-11

**Metareview:**

The paper proposes fairness-aware attention mechanism for contrastive learning, FARE and sparseFARE. While the motivation of the paper is clear -- to incorporate the fairness notion for contrastive learning -- the reviewers unanimously pointed out the paper needs to improve in terms of writing / clear comparison with previous work / and more extensive experiments on fairness benchmark datasets. We encourage the authors to revise the paper and submit again in the future venue.

**Justification For Why Not Higher Score:**

Experimental results as well as the exposition of the paper were weak to get accepted to ICLR.

**Justification For Why Not Lower Score:**

N/A

---

### Decision · Program_Chairs · 2024-01-16

Reject